# Simulated wind farm wake sensitivity to configuration choices in the Weather Research and Forecasting model version 3.8.1

**Jessica M. Tomaszewski**[1] **and Julie K. Lundquist**[1,2]

[1]Department of Atmospheric and Oceanic Sciences, University of Colorado, 311 UCB, Boulder, CO 80309, USA

[2]National Renewable Energy Laboratory, Golden, CO, USA

**Correspondence:** Jessica M. Tomaszewski (jessica.tomaszewski@colorado.edu)

**Abstract.** Wakes from wind farms can extend over 50 km downwind in stably stratified conditions. These wakes can undermine power production at downwind turbines, adversely impacting revenue. As such, wind farm wake impacts must be considered in wind resource assessments, especially in regions of dense wind farm development. The open-source Weather Research and Forecasting (WRF) numerical weather prediction model includes a wind farm parameterization to estimate wind farm wake effects, but model configuration choices can influence the resulting predictions of wind farm wakes. These choices include vertical resolution, horizontal resolution, and whether or not to include the addition of turbulent kinetic energy generated by the rotating wind turbines. Despite the sensitivity to model configuration, no clear guidance currently exists for these options. Here we compare simulated wind farm wakes produced by varying model configurations with meteorological observations near a land-based wind farm in flat terrain over several diurnal cycles. A WRF configuration comprised of horizontal resolutions of 3 km or 1 km paired with a vertical resolution of 10 m provides the most accurate representation of wind farm wake effects, such as the correct surface warming and elevated wind speed deficit. The inclusion of turbine-generated turbulence is also critical to produce accurate surface warming and should not be omitted.

*Copyright statement.* This work was authored [in part] by the National Renewable Energy Laboratory, operated by Alliance for Sustainable Energy, LLC, for the U.S. Department of Energy (DOE) under Contract No. DE-AC36-08GO28308. Funding provided by the U.S. Department of Energy Office of Energy Efficiency and Renewable Energy Wind Energy Technologies Office. The views expressed in the article do not necessarily represent the views of the DOE or the U.S. Government. The U.S. Government retains and the publisher, by accepting the article for publication, acknowledges that the U.S. Government retains a nonexclusive, paid-up, irrevocable, worldwide license to publish or reproduce the published form of this work, or allow others to do so, for U.S. Government purposes.

# 1   Introduction

Wind energy is growing rapidly to meet increasing energy demands with lower-carbon electricity sources. A wind turbine generates electricity by using momentum from the wind to turn its blades and generator, causing a downwind wake characterized by a reduction in wind speed and an increase in turbulence (Lissaman, 1979). The aggregate impact of these individual turbine wakes can extend over 50 km downwind of a wind farm, particularly during stable conditions when little atmospheric convection is present to erode the wake (Christiansen and Hasager, 2005; Platis et al., 2018). Consequences of these wake effects include local changes to surface fluxes that can raise surface temperatures at night caused by turbine-induced mixing of the nocturnal inversion (Baidya Roy, 2004; Baidya Roy and Traiteur, 2010; Zhou et al., 2012; Rajewski et al., 2013; Smith et al., 2013; Rajewski et al., 2016; Siedersleben et al., 2018a) and loss of power and revenue for downwind wind farms operating in the wind speed deficit (Nygaard, 2014; Nygaard and Hansen, 2016; Nygaard and Newcombe, 2018; Lundquist et al., 2018). As wind farms continue dense development, wind farm wake impacts must be considered in wind resource assessments.

Several numerical simulation tools exist to assess wind farm wake effects. Large-eddy simulations (LES) provide fine-scale information of near-turbine meteorological impacts of wind turbines (Sørensen and Shen, 2002; Vermeer et al., 2003; Calaf et al., 2010; Troldborg et al., 2010; Sanderse et al., 2011; Churchfield et al., 2012; Archer et al., 2013; Aitken et al., 2014; Mirocha et al., 2014; Abkar and Porté-Agel, 2015a, b; Vanderwende et al., 2016; Marjanovic et al., 2017; Tomaszewski et al., 2018). Simulating the turbine rotor and downstream flow with LES is useful, albeit computationally expensive, making realistic LES simulations of entire wind farms that can span 100s of km$^2$ unreasonable. Reynolds-averaged Navier-Stokes (RANS) approximations (Cabezón et al., 2011; Tian et al., 2014; Göçmen et al., 2016; Astolfi et al., 2018; Iungo et al., 2018) and industry flow models (e.g., FLOw Redirection and Induction in Steady State (FLORIS; NREL, 2019) and Wind Farm Simulator (WFSim; Boersma et al., 2016)) are commonly used for lower-cost wind farm wake investigations (Beaucage et al., 2012). However, these models are often limited in parameterizations of meteorological effects such as atmospheric stability or large-scale wind patterns.

One approach to parameterizing turbines numerically in simulations with grid spacings of kilometers or more is to exaggerate surface roughness to represent the local reduction of wind speed of wind farm wakes (Keith et al., 2004; Frandsen et al., 2009; Barrie and Kirk-Davidoff, 2010; Fitch, 2015). This enhanced surface roughness approach was later shown to produce erroneous predictions, including the wrong sign of surface temperature change through the diurnal cycle (Fitch et al.,

2013). Emeis and Frandsen (1993) proposed and later refined (Emeis, 2009) an analytical wind park model that considers both momentum loss and downward momentum flux, which accounts for the spatially averaged and stability dependent momentum-extraction coefficient by turbines. While the Emeis model incorporates the influence of additional wake characteristics, it lacks consideration for turbine-scale interactions between the rotor layer and the surface (Fitch et al., 2012; Abkar and Porté-Agel, 2015a).

Alternatively, the turbine power and thrust curves can define the elevated momentum sink and turbulence generation of a wind turbine. The turbine power and thrust curves give the relationship between hub-height inflow wind speed, power production, and force exerted onto the ambient air by a specific wind turbine type. The use of these turbine specifications can predict meteorological impacts of wind turbines at hub height extending down to the surface, forming the basis for multiple wind farm parameterizations in mesoscale numerical weather prediction models, such as the Wind Farm Parameterization (WFP) (Fitch et al., 2012; Fitch, 2016), the Explicit Wake Parametrisation (Volker et al., 2015), the Abkar and Porté-Agel (2015b) Parameterization, and a hybrid wind farm parametrization (Pan and Archer, 2018).

The open-source WFP of the Weather Research and Forecasting (WRF) model acts to collectively represent wind turbines in each model grid cell as a turbulence source and a momentum sink within the vertical levels of the turbine rotor disk (Fitch et al., 2012; Fitch, 2016). A fraction of the kinetic energy extracted by the virtual wind turbines is converted to power, reported as an aggregate sum in each model grid cell. The default setting of the WFP dictates that turbine-induced turbulence generation is derived from the difference between the thrust and power coefficients, though this option can be switched off to constrain turbulent kinetic energy (TKE) to be added only via wind shear arising from the momentum deficit in the wake of the turbine. Wind-speed-dependent thrust coefficients specify the local wind drag on kinetic energy extraction as well as on power estimation. Users can modify the specifications of the parameterized turbine, such as its hub height, rotor diameter, power curve, and thrust coefficients, as well as its latitude and longitude location.

The WRF WFP has been employed in many studies with different model configurations to assess the impacts of onshore and offshore wind farms (e.g., Eriksson et al. 2015; Jimenez et al. 2015; Miller et al. 2015; Vanderwende et al. 2016; Vanderwende and Lundquist 2016; Wang et al. 2019). WFP simulations have reproduced the observed localized, nighttime, near-surface warming caused when wind turbines mix warmer air from the nocturnal inversion down to the surface (Fitch et al., 2013; Lee and Lundquist, 2017b; Siedersleben et al., 2018a; Xia et al., 2019). Such findings have prompted additional studies using the WFP to address whether large-scale wind farms could alter regional climate (see Table 1 for an overview). Specific examples include Vautard et al. (2014), who use WFP simulations of future European deployment at a 50-km horizontal resolution and ~30-m vertical resolution to find statistically significant temperature signals only in winter, constrained to ±0.3 K. Conversely, Pryor et al. (2018) use the WRF WFP at a 4-km horizontal resolution and ~30-m vertical resolution to find that wind farm-induced surface warming (<0.1 K on average) around wind farms in Iowa is more significant during summer months. Miller and Keith (2018), using WFP simulations at a 30-km horizontal resolution and 25-m vertical resolution, suggest that generating

**Table 1.** Overview of Previous WRF WFP Wake Impact Studies

| Reference | Horizontal resolution[1] | Vertical resolution[2] | Near-surface T impact due to vertical redistribution of heat[3] |
|---|---|---|---|
| Fitch et al. (2013) | 1 km | 15 m | increase of 0.5 K |
| Vautard et al. (2014) | 50 km | 30 m | wintertime changes $\pm 0.3$ K |
| Miller and Keith (2018) | 30 km | 25 m | increase of 0.24 K across CONUS |
| Pryor et al. (2018) | 4 km | 30 m | increase of <0.1 K |
| Siedersleben et al. (2018a) | 1.67 km | 35 m | increase of 0.4 K |
| Wang et al. (2019) | 1 km | 7 m | wintertime increase of 0.2 K |
| Xia et al. (2019) | 1 km | 20 m | increase of 0.3 K |

[1] Of innermost domain, if applicable
[2] Within rotor layer
[3] Within wind farm, unless otherwise specified

today's United States electricity demand (which they estimate to be 0.46 $TW_e$) with only wind power would redistribute boundary-layer heat to warm the continental United States (CONUS) surface temperatures by 0.24 K.

The varying WRF WFP configurations employed in previous wake studies present conflicting depictions of the impact of wind farm wakes, suggesting a sensitivity to model settings. Past studies have begun evaluating this sensitivity in the WRF WFP. Lee and Lundquist (2017a) note that a ∼12-m vertical resolution is necessary to reproduce observed power production, while Mangara et al. (2019) find that the wake dynamics simulated by the WFP are more sensitive to horizontal resolution than vertical resolution. Xia et al. (2019) find differences in the WFP solution of surface temperature changes, depending on the inclusion of the turbine-generated TKE term. Siedersleben et al. (2019) determine that a TKE source and a horizontal resolution on the order of 5 km or finer are necessary to represent the impact of offshore wind farms on the stably stratified, marine atmospheric boundary layer. Sensitivity studies conducted for the New European Wind Atlas find a sensitivity to modifications to the MYNN scheme in different WRF versions (Witha et al., 2019; Hahmann et al., 2020). The MYNN scheme within WRF Versions 3.7.X to 3.9.X differs from 4.X most notably in the drag coefficient parameterization in the surface layer subroutine and the mixing length formulation, leading to differences in the wind that could impact wake studies (Olson et al., 2016). While these studies give initial guidance on the sensitivity of the WFP to model settings, a greater breadth of spatial resolution and WFP TKE sensitivity tests are needed to formulate more confident best practice guidelines.

Here we expand upon and synthesize the work of Lee and Lundquist (2017a), Mangara et al. (2019), Xia et al. (2019), and Siedersleben et al. (2019) and provide guidance on optimal WRF WFP model settings for simulating wakes. Section 2 outlines the model setup and configurations tested, Section 3 presents the differences in the configurations, and Sections 4 and 5 summarize our results confirming model sensitivity and recommend WRF WFP modeling choices.

**Table 2.** Simulation Configurations

| Identifier | Horizontal resolution (dx) | Vertical resolution (dz) | Time step (dt) | TKE option | WRF version | Computation time (CPU hrs)[1] |
|---|---|---|---|---|---|---|
| dx03_dz10_dt30_tke | 3 km | 10 m | 30 s | Default | 3.8.1 | 200 |
| dx09_dz10_dt30_tke | 9 km | 10 m | 30 s | Default | 3.8.1 | 50 |
| dx27_dz10_dt30_tke | 27 km | 10 m | 30 s | Default | 3.8.1 | 12 |
| dx03_dz30_dt30_tke | 3 km | 30 m | 30 s | Default | 3.8.1 | 150 |
| dx03_dz10_dt30_ntke | 3 km | 10 m | 30 s | No added TKE | 3.8.1 | 200 |
| dx03_dz10_dt10_tke | 3 km | 10 m | 10 s | Default | 3.8.1 | 650 |
| dx01_dz10_dt30_ntke | 1 km | 10 m | 30 s | No added TKE | 3.8.1 | 630 |
| dx01_dz10_dt30_tke | 1 km | 10 m | 30 s | Default | 3.8.1 | 630 |
| dx03_dz10_dt30_tke_V4 | 3 km | 10 m | 30 s | Default | 4.0 | 200 |

[1] Per 1 day (24 hours + 12 hour spinup) of simulation. Domain sizes indicated in Fig. 1.

## 2 Data and methods

### 2.1 Observations

This sensitivity analysis relies on data from the Crop Wind Energy Experiment (CWEX). CWEX investigated the intersection of agriculture and wind energy within the planetary boundary layer (Rajewski et al., 2013). The site was characterized by generally flat terrain and a vegetated surface of corn and soybeans and featured an operating wind farm northeast of Ames, Iowa. In the 2013 CWEX campaign, seven surface flux stations, a radiometer, three profiling lidars, and a scanning lidar were deployed within and around this wind farm to explore the interaction of multiple wakes in a range of atmospheric stability conditions (Vanderwende et al., 2015; Bodini et al., 2017; Sanchez Gomez and Lundquist, 2019).

The WINDCUBE 200S scanning lidar was positioned within the northern half of the wind farm during CWEX-13, about 6 rotor diameters north of the nearest turbine row. The 200S lidar utilized a velocity azimuth display (VAD) scanning strategy that measured winds from ∼100 m to 4,800 m above ground level (AGL) approximately every 50 m. We use the 200S 75° elevation scans (Vanderwende et al., 2015) in this study to estimate horizontal winds every 30 minutes to validate the boundary-layer winds simulated by WRF.

We select August 24 through 27 of 2013 for our analysis. During this period, a lack of major synoptic events allowed strong nightly nocturnal low-level jets (LLJ) to occur (Vanderwende et al., 2015). Lee and Lundquist (2017a) found that WRF performed well in capturing the timing, intensity, and position of these low-level jets. Furthermore, the wind turbines operated without curtailment, and the instruments were online collecting data, making this period ideal for a model sensitivity and performance evaluation.

### 2.2 Modeling

We conduct all simulations but one in our sensitivity study with version 3.8.1 of the Advanced Research WRF (ARW) model (Skamarock and Klemp, 2008). While model time step, vertical resolution, horizontal resolution (and thus domain size), and

model version are among the model settings varied to test sensitivity, several model options are kept consistent across all simulations based on previous studies of this time period. The 0.7° ERA-Interim (ECMWF, 2009; Dee et al., 2011) data set provides initial and boundary conditions for all model runs, chosen for its better performance over other reanalysis data sets (Lee and Lundquist, 2017a; Hahmann et al., 2020). Topographic data are provided at 30-s resolution. Physics options include the Rapid Radiative Transfer Model (RRTM) long-wave radiation scheme (Mlawer et al., 1997), the single-moment 5-class microphysics scheme (Hong et al., 2004), land surface physics with the Noah Land Surface Model (Ek et al., 2003), Dudhia short-wave radiation (Dudhia, 1989) with a 30-s time step, a surface layer scheme that accommodates strong changes in atmospheric stability (Jimenez et al., 2012), the second order Mellor-Yamada-Nakanishi-Niino (MYNN2) PBL scheme (Nakanishi and Niino, 2006) *without* TKE advection, and the explicit Kain–Fritsch cumulus parameterization (Kain, 2004) on domains with horizontal resolutions coarser than 3 km.

We simulate each 24-hour day of August 24 through 27 individually, beginning spinup at 1200 Coordinated Universal Time (UTC) on the previous day with analysis retained after 0000 UTC. We define the wake effect by comparing a simulation without the WFP to a simulation with the WFP, as in Fitch et al. (2012), Lee and Lundquist (2017a), and Redfern et al. (2019). We use the power and thrust curve of the 1.5-MW Pennsylvania State University (PSU) generic turbine (Schmitz, 2012) to parameterize the wind turbines, based on the General Electric SLE turbine (80-m hub height and 77-m rotor diameter). This turbine model closely matches those installed in the wind farm present at the CWEX site, and Siedersleben et al. (2018b) show little sensitivity to the exact turbine power curve for similar turbines. For turbines with substantially different ratings, the exact power curves should be used.

We define a "baseline" configuration (*dx03_dz10_dt30_tke*) around which we modify various settings. This baseline is set to have three nested domains with horizontal resolutions of 27, 9, and 3 km, respectively, where the innermost, 3-km domain (*dx03*) covers the state of Iowa, centered over the simulated wind farm (Fig. 1). The vertical resolution of the baseline is nominally defined to be ∼10 m in the lowest 200 m (*dz10*), stretching vertically thereafter. The model time step is 30 s on the outer domain (*dt30*), reducing by a factor of 3 for each additional nest. Turbine-induced turbulence is parameterized via an addition of TKE (*tke*), the default WFP option. We then vary the horizontal resolution (*dx*), vertical resolution (*dz*), time step (*dt*), turbulence option (*tke* or *ntke*), and WRF model version, 3.8.1 vs 4.0 (*V4*), about this baseline configuration to make up our sensitivity test (Table 2).

For example, we test vertical resolution sensitivity by comparing the baseline configuration to the *dx03_dz30_dt30_tke* configuration, which coarsens the vertical resolution from 10 m in the baseline to 30 m (*dz30*) and reduces the number of layers intersecting the turbine rotor layer from ∼7 to 3 (Fig. 2). We test sensitivity to horizontal resolution by separately nesting higher resolution domains, first using only a 27-km domain, then adding a 9-km domain, a 3-km domain, and finally a 1-km domain (Fig. 1). Our finest domain tested is 1 km to avoid issues with the Terra Incognita (Wyngaard, 2004; Ching et al., 2014; Zhou et al., 2014; Doubrawa and Muñoz Esparza, 2020). Additionally, we test the impacts of the WFP turbine-generated TKE source by running two simulations with this option switched off (*dx03_dz10_dt30_ntke* and *dx01_dz10_dt30_ntke*). Disabling

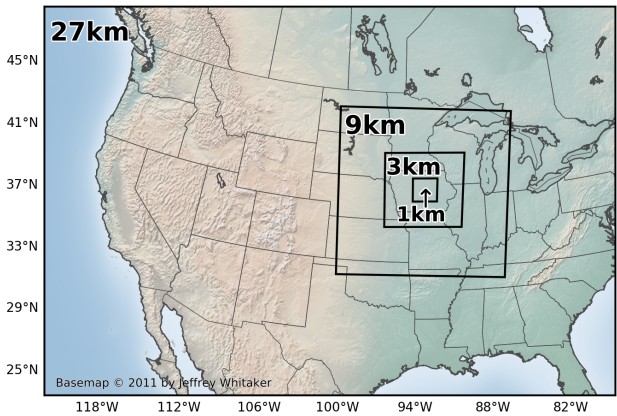

**Figure 1.** Map representing the domains (starting at 27 km, nesting down to 9 km, 3 km, and 1 km) of the horizontal resolution tests that also serve as the outer domains for finer resolutions. Geography data provided by Matplotlib's (Hunter, 2007) Basemap © 2011 by Jeffrey Whitaker.

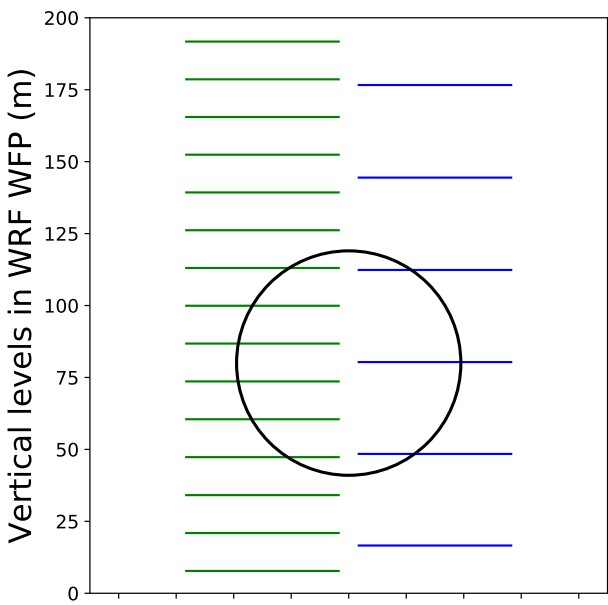

**Figure 2.** Schematic of the two vertical grids tested and where they typically intersect the turbine rotor layer (black circle): the ∼10-m grid (*dz10*) on the left in green and the ∼30-m grid (*dz30*) on the right in blue

the TKE generation is done by commenting out line 226 (the qke(i,k,j) calculation) in module_wind_fitch.F and recompiling WRF. We examine sensitivity to model time step by running a simulation at a time step of 10 s on the outermost domain (*dx03_dz10_dt10_tke*), refined from 30 s in the other configurations. Finally, we assess sensitivity to WRF version by running a simulation with the same configuration as the baseline (*dx03_dz10_dt30_tke*) with version 4.0 of WRF. Sensitivities of the tested configurations are determined via comparisons of model solutions of the wind farm wake, including the area of wake coverage and the magnitude of hub-height wind speed deficits and near-surface temperature changes. All simulation configurations are outlined in Table 2.

## 3   Results

### 3.1   Performance of non-WFP WRF

We first verify that the WRF simulations without the WFP, i.e., "no wind farm" (NWF) simulations, simulate accurate ambient winds compared to the CWEX scanning lidar measurements collected from outside the wind farm. Qualitatively, all WRF configurations in our sensitivity test have skill in simulating the timing and position of the LLJ (a similar finding of Smith et al. (2019)) but overestimate the magnitude of the LLJ wind speed increase (Fig. 3) and predict fewer occurrences of easterly winds, especially on Aug 24 and 25 (Fig. 4, not all configurations shown). We include time series of wind speed from two simulations, *dx27_dz10_dt30_tke* (Fig. 3a, Fig. 4a) and *dx01_dz10_dt30_tke* (Fig. 3b, Fig. 4b), as an example. The finer-horizontal resolution NWF *dx01_dz10_dt30_tke* better captures the intermittency in strength of the LLJ (Fig. 3a), though both simulations overestimate wind speed compared to the scanning lidar. Comparisons of the simulated near-hub-height hourly wind speed and direction against the scanning lidar further illustrate the positive wind speed bias (Fig. 5a) and more westerly wind direction bias (Fig. 5b) by the simulations, with the 27-km horizontal resolution simulation showing higher biases than the 1 km in both cases. The RMSE differences between the 27-km and 1-km configurations are small in the wind direction estimates ($41.8°$ vs. $41.4°$, respectively), but differ more in the wind speed estimates, with the 27-km configuration exhibiting an RMSE of 3.1 m s$^{-1}$ compared to the 1-km RMSE of 2.8 m s$^{-1}$. Evaluation of other heights (150 m and 200 m, not shown) reveal a similar pattern of higher biases in the 27-km domain than the 1-km, particularly in wind speed.

Our comparisons of NWF simulations and scanning lidar measurements are similar to those of Lee and Lundquist (2017a), who also found good agreement in the occurrence of the LLJ. Lee and Lundquist (2017a) noted slightly better agreement in LLJ strength between simulations and the scanning lidar (i.e., an absolute error on the order of 1 m s$^{-1}$), which may be related to the larger domain sizes of their simulations.

### 3.2   WRF WFP sensitivity to model settings

#### 3.2.1   Impact on hub-height wind speed deficits

For an initial qualitative assessment of WRF WFP sensitivity to model settings, we compare snapshots of the wake effect at a single point in time among the various model configurations by subtracting the NWF simulation from the WFP simulation. We select 0200 UTC of Aug 26 (2100 Aug 25 local time) to examine because of the presence of strong southwesterly LLJ winds within and above the turbine rotor layer and the easily discernible wake impacts across all configurations tested, though many other time periods could have provided a similarly qualitative comparison.

The baseline simulation on 0200 Aug 26 (Fig. 6a) shows a clear hub-height wind speed deficit downwind of the wind farm, its impact extending over 40 km downwind with a maximum wind speed deficit close to 1.5 m s$^{-1}$. Changing the horizontal resolution of the WRF WFP reveals a clear sensitivity. Configurations with a coarser horizontal grid spacing predict wind speed

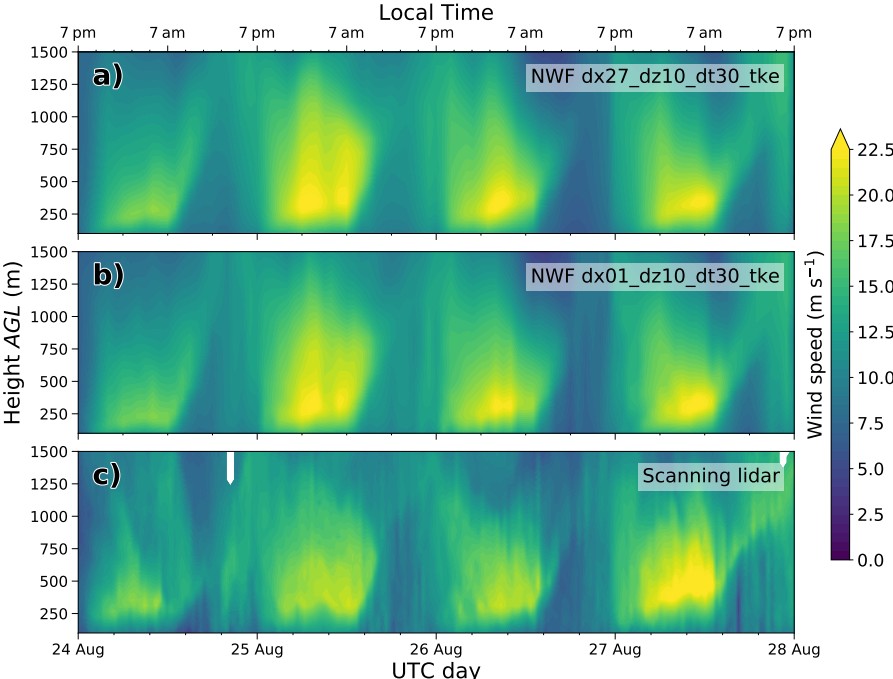

**Figure 3.** Time-height cross sections comparing wind speed from a) the no wind farm (NWF) run of the *dx27_dz10_dt30_tke* simulation with b) the NWF run of *dx01_dz10_dt30_tke*, and c) the scanning lidar observations

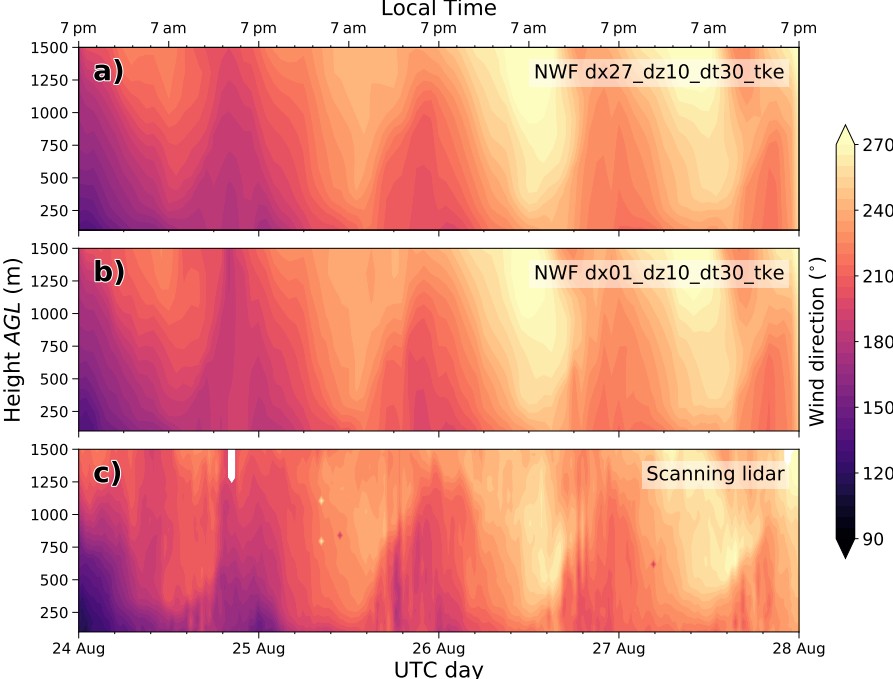

**Figure 4.** Time-height cross sections comparing wind direction from a) the no wind farm (NWF) run of the *dx27_dz10_dt30_tke* simulation with b) the NWF run of *dx01_dz10_dt30_tke*, and c) the scanning lidar observations

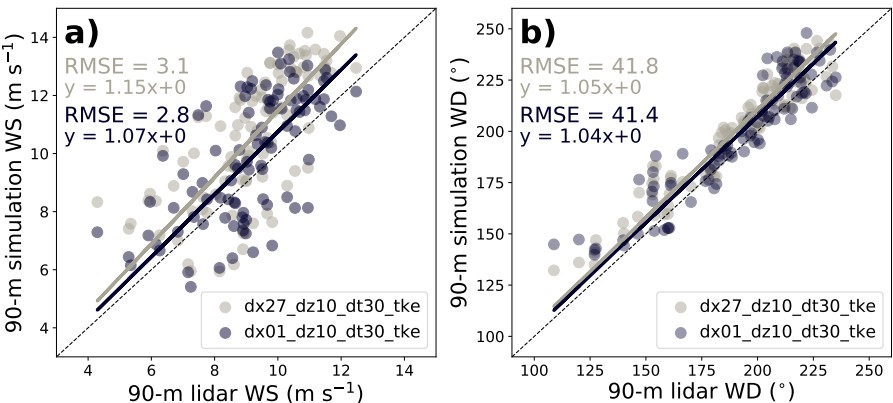

**Figure 5.** Hourly values of 90-m a) wind speed (WS) and b) wind direction (WD) from two of the simulations tested plotted against those of the scanning lidar. Dots, lines of best fit, and their corresponding equations and RMSE calculations in both panels are colored based on the model configuration denoted in the legend. One-to-one lines are dashed in black.

deficits smaller in magnitude than the baseline, but spanning a larger area (Fig. 6a,b). Conversely, a finer horizontal resolution (Fig. 6h) reduces the area of impact but increases the magnitude of the wind speed deficit. Coarsening the near-surface vertical resolution from 10 m in the baseline simulation to 30 m impacts the model solution by changing the shape of the wind speed deficit region (Fig. 6d). Disabling the WFP turbine-generated TKE option also only slightly impacts the shape of the wind

5 speed deficit, although it has negligible impacts on its magnitude (Fig. 6e), regardless of horizontal grid spacing (Fig. 6g,h). Using WRF version 4.0 (Fig. 6i) also creates subtle differences in the shape of the far wake and in the magnitude of the deficit in the near wake.

To explicitly quantify differences between the tested configurations, we sum the total area impacted by a particular magnitude of waking impact (e.g., wind speed deficit) in hourly increments. For example, the area impacted by an 80-m wind speed deficit

10 of 1 m s$^{-1}$ in the baseline simulation on Aug 26 at 0200 UTC (Fig. 6a) is calculated to be about 80 km$^2$ (denoted at the dashed black line in Fig. 7). We repeat the calculation for several defined deficits of interest for each hour in the period for all simulations (Fig. 7).

This time series of wake impact areas gives insight on the temporal variability of waking, as we see the largest areas of impact (larger dots in Fig. 7) and the strongest magnitudes of deficit occur during the night, with little to no wake impact

15 areas present during the day when increased ambient turbulence erodes wakes, similar to the stability dependence highlighted in Lundquist et al. (2018). As expected, all simulations exhibit larger areas impacted by lower deficit magnitudes (0.4−0.6 m s$^{-1}$) relative to instances of stronger deficits (>1 m s$^{-1}$). Clear differences emerge in the details of wake area coverage between the model configurations, especially when horizontal resolution is varied (Fig. 7a). As the horizontal resolution is coarsened from 1 km (dark purple), to 3 km (green), to 9 km (pink), and finally to 27 km (gray), the simulation increasingly

20 fails to capture higher-magnitude wake impacts and instead predicts larger areas of minor deficits.

The spatial extent of wind speed deficit impact appears to be most sensitive to horizontal resolution and is less sensitive to the other model settings tested (Fig. 7b). Holding horizontal resolution constant and reducing the model time step (blue)

or disabling the WFP turbulence generation (light purple) does not cause significant changes to the simulations' wind speed deficit extent across the range of impact magnitudes examined. The simulation with coarse vertical grid spacing (yellow) differs most from the others in Fig. 7b, predicting slightly larger regions of relatively weaker wake impacts, following the trend of the coarser horizontal resolution simulations. The newer version of WRF predicted a similar time series of deficits as the baseline (green) but was omitted from Fig. 7 to reduce clutter.                                                                                                           5

We next integrate the areas impacted by each defined deficit value in time across the full period (Fig. 8) to corroborate earlier suggestions of sensitivity to model configuration in Figs. 6 and 7. The coarser horizontal grid spacing simulations predict the largest overall region, with weaker wind speed deficits than the finer grid spacing simulations. Additionally, the coarse vertical resolution simulation produces smaller regions of strong wake impacts ($>1.6$ m s$^{-1}$ deficits) than its finer vertical resolution counterparts. Throughout the range of waking magnitudes, the original baseline simulation (3-km horizontal resolution, green)     10 closely matches the fine-resolution (1 km) simulation's estimates of wake coverage, suggesting WRF WFP estimates of wake impact and spatial coverage begin to converge at 3-km horizontal grid spacing (Fig. 8). Disabling the WFP TKE term has minimal impact between the two 1-km (red vs. dark purple) and two 3-km (light purple vs. green) simulation pairs examined when considering wind speed deficit effects. Differences between the baseline's 30-s time step and the reduced, 10-s time step simulation (blue) are small. The baseline case run with version 3.8.1 and the case run with version 4.0 (green vs. orange)     15 predict similar total waking impacts over the period, deviating most ($\sim$100 km$^2$) at the 1.8-m s$^{-1}$ deficit (Fig. 8).

To supplement Fig. 8, we next compare the average wake effects predicted by the different simulation cases (Fig. 9). As previously noted, the coarser horizontal resolution simulations predict the largest average affected regions, with weaker maximum wind speed deficits than the finer resolution simulations. Differences between average wake impact predicted by the other configurations are more subtle, with those run at a 30-m vertical resolution or lacking the WFP TKE term deviating most from     20 the baseline (green). Subtle sensitivity exists to the model time step and version, most apparent in the average areas impacted by the strongest deficits, i.e., 1.8 and 2.0 m s$^{-1}$ (Fig. 9). Such large deficits occur more infrequently than others, meaning averages could exaggerate the differences between configurations there.

### 3.2.2   Impact on near-surface temperature and moisture changes

Another wind farm wake effect sensitive to WRF WFP model settings is the presence and sign of near-surface temperature     25 changes. As with the wind speed deficit analysis (Fig. 6), initial comparisons of snapshots of the temperature changes on Aug 26 at 0200 UTC reveal differences between the configurations (Fig. 10, 11). The baseline simulation shows a clear nighttime warming signal at 2 m in the immediate vicinity of the wind turbines (Fig. 10a), consistent with satellite observations (Zhou et al., 2012) and in-situ observations (Rajewski et al., 2013). This warming is caused by a redistribution of heat mixed down from hub height (Fig. 11a), as shown by a concurrent vertical slice transecting south-north through the wind farm (dashed line     30 in Fig. 10a). Changing the horizontal resolution of the WRF WFP has a notable impact on the spatial coverage of the near-surface warming. Simulations with coarser horizontal resolutions predict weaker 2-m temperature warming signals that span

greater areas (Fig. 10b,c; 11b,c), which parallel results from the wind speed deficit analysis (Fig. 6b,c). Similarly, reducing the time step or using WRF version 4.0 has little impact on the model solution of temperature changes based on the snapshots from Fig. 10f,i and Fig. 11f,i.

Changes to turbine-generated turbulence and vertical resolution exert the greatest impacts on model solutions of temperature signals. Coarsening the vertical grid spacing from 10 m to 30 m reverses the sign of the near-surface temperature change, producing an unphysical localized region of surface cooling in the immediate vicinity of the wind farm (Fig. 10d, 11d). Similarly, disabling turbine-generated turbulence also causes a cooling signal near the surface, irrespective of horizontal resolution (Fig. 10e,g; 11e,g). Temperature profiles from these configurations reveal slight warming within grid cells at turbine hub height that does not mix down to the surface (Fig. 11d,e,g). This cooling signal produced by simulations with too-coarse vertical resolution or lacking turbine-generated TKE directly conflicts with wind farm wake observations of localized near-surface warming during stable conditions (e.g., Zhou et al. 2012; Rajewski et al. 2013). All configurations produce cooling just above turbine hub height and warming within the rotor layer, illustrating the redistribution of heat that occurs from mixing of the nocturnal inversion; however, sufficient vertical resolution and turbine-generated turbulence is required to mix warmer temperatures down to the surface (Fig. 11).

We sum at hourly increments the total area impacted by a particular magnitude of near-surface temperature change to explicitly quantify differences between tested configurations (Fig. 12). We limit the bounding area of interest to grid cells immediately around the wind farm to consider the more localized nature of near-surface temperature impacts, as opposed to the larger downwind fetch impacted by a wind speed deficit considered in Fig. 7. The temporal variability of waking again appears, with relatively larger areas of temperature impacts occurring overnight, typically emerging as a warming signal. These temperature impacts are clearly sensitive to horizontal grid spacing throughout the time period (Fig. 12a). As the horizontal resolution is coarsened from 1 km (dark purple), to 3 km (green), to 9 km (pink), and finally to 27 km (gray), the simulation predicts larger areas of minor deficits and fails to capture higher-magnitude temperature increases. Occasional instances of cooling occur typically just before sunset and are exaggerated in coarser horizontal resolution configurations, likely caused by convection in the daytime with locations shifted due to the presence of the wind farm.

Coarsening vertical resolution to 30 m (yellow) or disabling TKE generation (light purple) incorrectly produces a nocturnal cooling signal across the time period (Fig. 12b), most notably on Aug 25 through 27. WRF WFP needs to be able to resolve wind shear to vertically mix warmer inversion air to the the surface as documented in observations (e.g., Rajewski et al. 2013; Smith et al. 2013; Rajewski et al. 2016; Platis et al. 2018; Siedersleben et al. 2018a), and these configurations with too-coarse vertical grid spacing (30 m) or lacking turbine-generated TKE clearly fail to generate such wind shear throughout most of the period. An exception occurs on Aug 24, when more southeasterly winds (Fig. 4) aligning with the orientation of the wind farm cause a narrow, highly concentrated wake region that permits the 30-m vertical resolution configuration to produce a surface warming signal. While WRF WFP wake effects experience strong sensitivity to vertical resolution and TKE

generation, reducing the model time step (blue) from the baseline (green) has little impact on the temperature change solution throughout the period (Fig. 12b).

As with the wind speed deficit analysis, we next integrate (Fig. 13) and average (Fig. 14) the areas impacted by each defined deficit value in time across the full period, which reiterates that significant model sensitivities exist. The configurations without turbine-generated turbulence at both 1-km and 3-km horizontal resolutions (red, purple, respectively) or with a coarse, 30-m vertical resolution (yellow), exhibit the largest erroneous overall areas of significant cooling signals in the vicinity of the wind farm (Fig. 13, 14). The 9-km horizontal resolution (pink) also experiences relatively stronger cooling impacts across the period, but estimates total areas impacted by warming to span hundreds of kilometers more. Other configurations that experience cooling with adequate (10-m) vertical grid spacing and the turbine-TKE enabled estimate such cooling to be minimal in total coverage and magnitude (Fig. 13, 14).

All configurations predict some warming immediately around the wind farm throughout the period (Fig. 13, 14), while those with the 30-m vertical grid or lacking turbine-generated TKE produce the smallest areas of warming. Configurations with the coarsest horizontal resolutions (27 km, grey; 9 km, pink) predict large areas of impact by weak warming signals. Only configurations with a 3-km or finer horizontal grid are able to capture warming impacts above 0.4 K. Reducing model time step (blue) or using version 4.0 of WRF (orange) again has little impact on the overall prediction of temperature impact coverage compared to the baseline (green) (Fig. 13, 14).

Another impact of wind farm wakes is the changes to the near-surface moisture content, when can happen overnight when the enhanced mixing from the wind farm brings relatively drier air down and moister air up, leading to a drying near the surface and moistening aloft (Baidya Roy, 2004; Siedersleben et al., 2018a). We examine the model sensitivity in producing this moisture effect via a vertical snapshot of water vapor mixing ratio (q) changes on Aug 26 at 0200 UTC (Fig. 15). As with the temperature analysis (Fig. 11), changing the horizontal resolution of the WRF WFP has a notable impact on the spatial coverage and intensity of the near-surface drying. Simulations with coarser horizontal resolutions predict weaker near-surface drying signals that span larger areas (Fig. 15b,c).

However, changes to turbine-generated turbulence and vertical resolution have the greatest impacts on model solutions of moisture signals. Coarsening the vertical grid spacing from 10 m to 30 m reverses the sign of the moisture change, producing a localized region of surface moistening in the immediate vicinity of the wind farm (Fig. 15d). Similarly, disabling turbine-generated turbulence also causes an increase in water vapor near the surface, irrespective of horizontal resolution (Fig. 15e,g). This moistening signal produced by simulations with too-coarse vertical resolution or lacking turbine-generated TKE contradicts observations of localized near-surface drying during stable conditions (Baidya Roy, 2004; Siedersleben et al., 2018a), implying that those simulations lack sufficient mixing, the same deficiency that produces erroneous cooling signals (Fig. 11) as well. We omit discussing the model sensitivity in producing moisture impacts with the same detail as the temperature changes, as the moisture impact results parallel those of the temperature impacts.

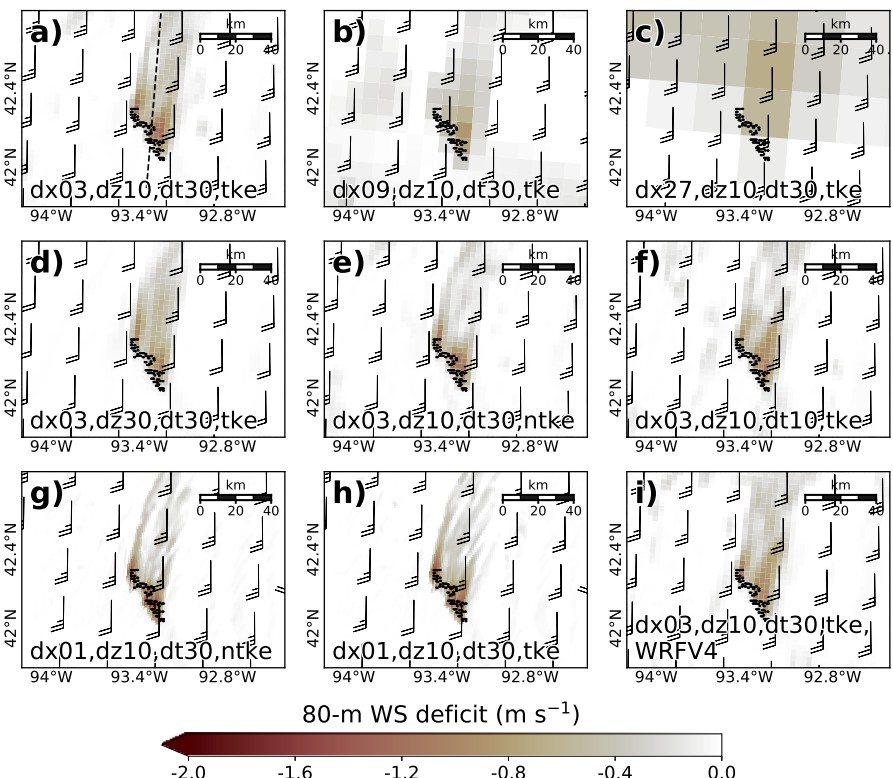

**Figure 6.** Hub-height (∼80 m) wind speed (WS) deficits resulting from the presence of the wind farm in the tested simulations on Aug 26 0200 UTC (Aug 25 2100 LT). The 80-m wind barbs from the wind farm simulation are plotted in knots every 27 km, regardless of horizontal resolution. The dashed line in panel *a* denotes the location of the vertical cross section in Fig. 11. Panels are cropped to the same region around the wind farm despite certain configurations having varying simulation domains depending on horizontal resolution.

## 4    Discussion

We compare different WRF WFP simulation solutions of land-based wind farm wake effects in simple terrain and meteorological conditions to quantify the sensitivity of the simulations to model configuration and thereby define recommendations for best-practice model settings. Settings tested include horizontal and vertical grid spacing, model time step, model version, and inclusion of turbine-generated turbulence. We divide our analysis into the two main atmospheric impacts of a wind farm wake: a hub-height wind speed deficit extending downwind of the wind farm and a nighttime near-surface temperature increase immediately around the wind farm mixed down from the nocturnal inversion.

In summary, simulated WRF WFP solutions of wind speed deficits are most sensitive to the horizontal resolution. Horizontal grids of 3 km and 1 km converged on similar depictions of the magnitude and spatial coverage of wind deficits, while grids of 9 km or larger dilute the wake impact over large areas. Solutions of 2-m temperature (and similarly moisture) changes are also sensitive to horizontal resolution in that too-coarse (>9 km) grid spacing results in large expanses of weak temperature changes, contradicting the typically localized nature of temperature impacts from a wind farm seen in observations (e.g., Baidya Roy 2004; Zhou et al. 2012; Rajewski et al. 2013; Siedersleben et al. 2018a). The more important model settings to consider for accurate representation of surface temperature impacts of wind farm wakes, however, are the vertical resolution

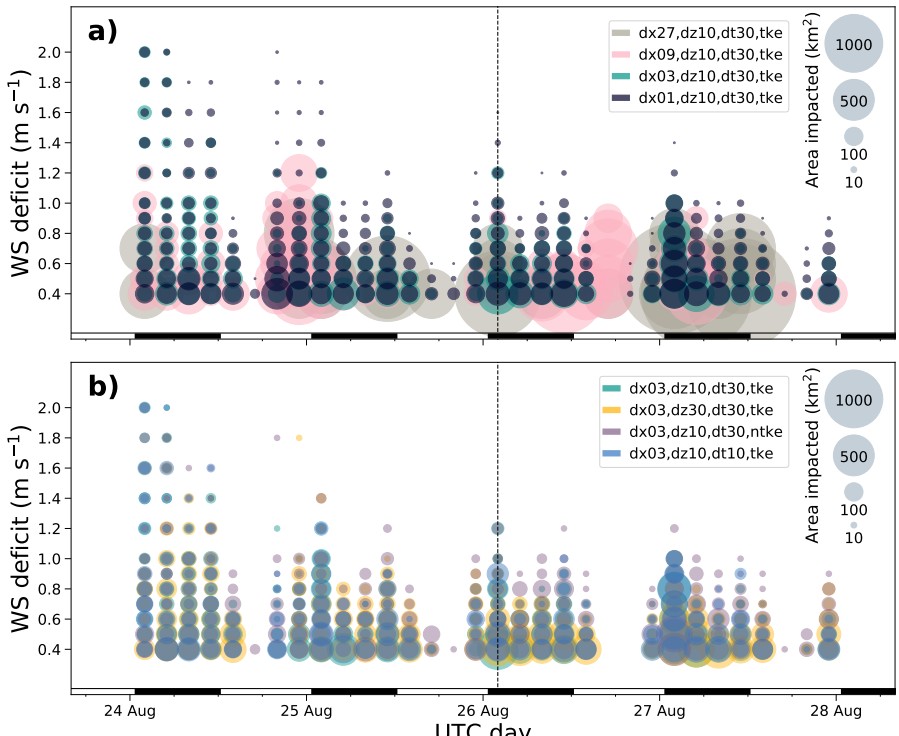

**Figure 7.** Time series of area impacted by the wake-induced, 80-m wind speed deficits as predicted by the tested simulations, plotted every 3 hours throughout the period. The size of the dots represents the spatial coverage of their respective magnitude of impact (scale denoted in top right), with each dot colored based on its configuration. Areas of impact were calculated for deficits every 0.1 m s$^{-1}$ between 0.4 and 1.0 m s$^{-1}$, then every 0.2 m s$^{-1}$ until 2.0 m s$^{-1}$. The configurations are divided into groups that (a) vary horizontal resolution and (b) hold horizontal resolution constant and vary other model settings. The time chosen for the qualitative analysis (Aug 26 0200 UTC) in Fig. 6 is denoted by the black dashed line. Black and white bar at bottom denotes post-sunrise (white) and post-sunset (black).

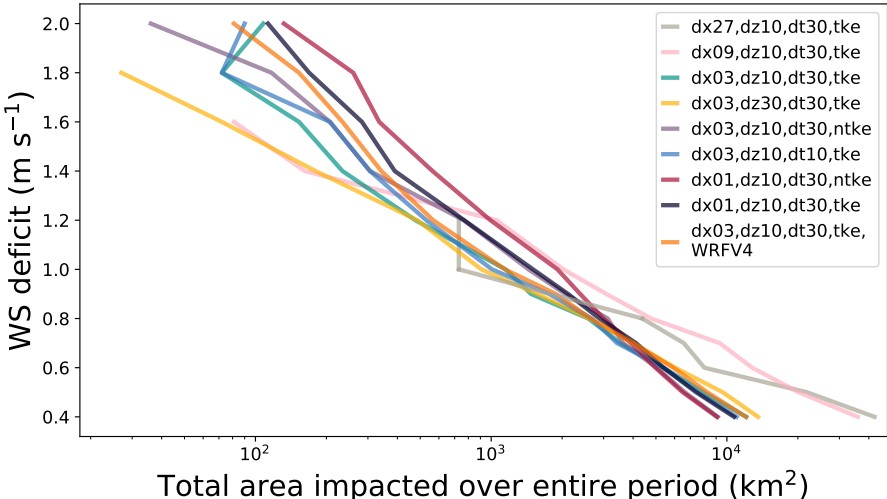

**Figure 8.** Total area impacted by the wake-induced, 80-m wind speed deficits as predicted by the tested simulations, plotted at different magnitudes of impact and integrated in time across the entire period. Each line is colored based on its configuration.

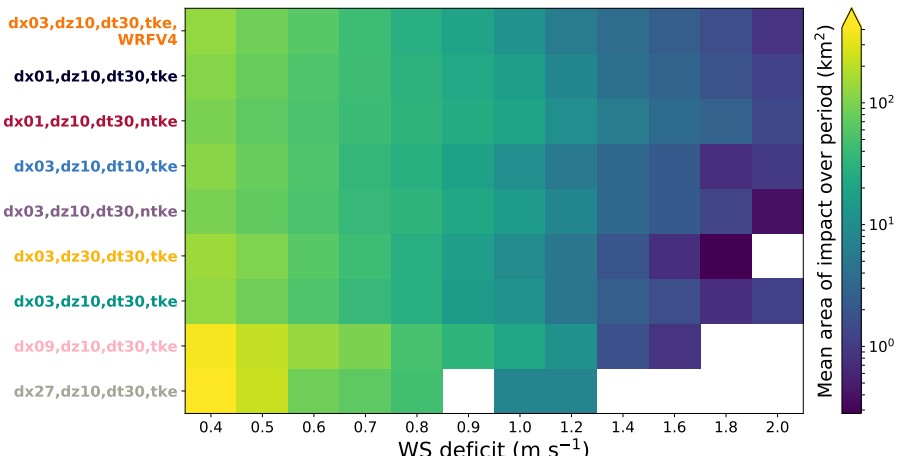

**Figure 9.** Average area affected by each 80-m wind speed deficit (columns) over the entire simulation period for each simulation tested (rows). Yellow squares indicate larger areas of impact, with empty (white) squares indicating a lack of occurrence for that particular magnitude of impact.

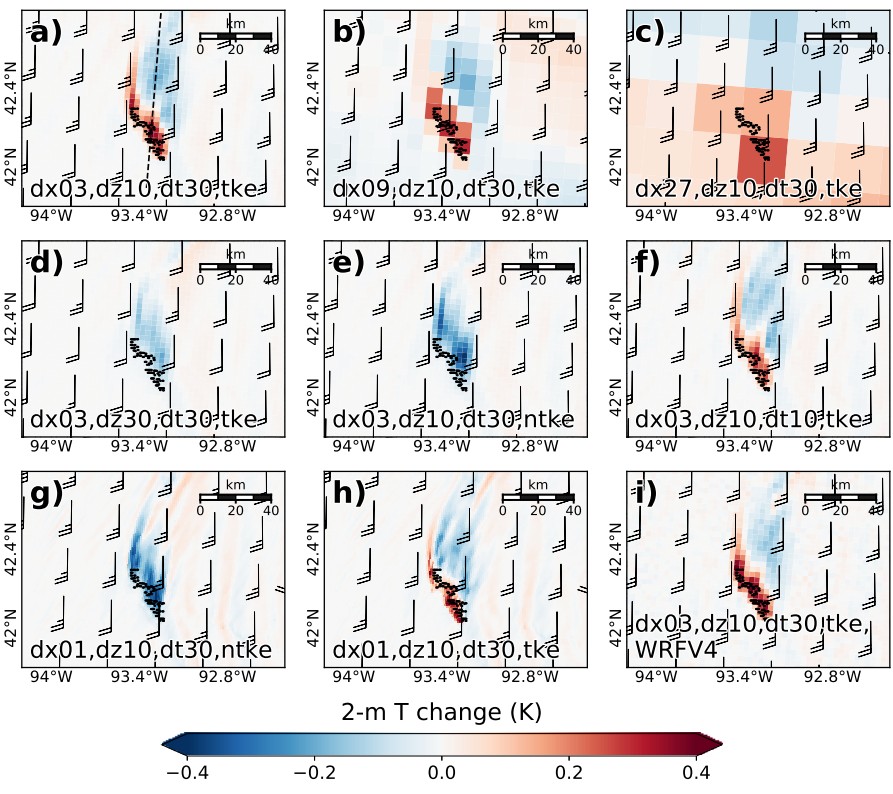

**Figure 10.** Near-surface (2 m) temperature (T) changes resulting from the presence of the wind farm in the tested simulations on Aug 26 0200 UTC (Aug 25 2100 LT). The 80-m wind barbs from the wind farm simulation are plotted in knots every 27 km, regardless of horizontal resolution. The dashed line in panel *a* denotes the location of the vertical cross section in Fig. 11. Panels are cropped to the same region around the wind farm despite certain configurations having varying simulation domains depending on horizontal resolution.

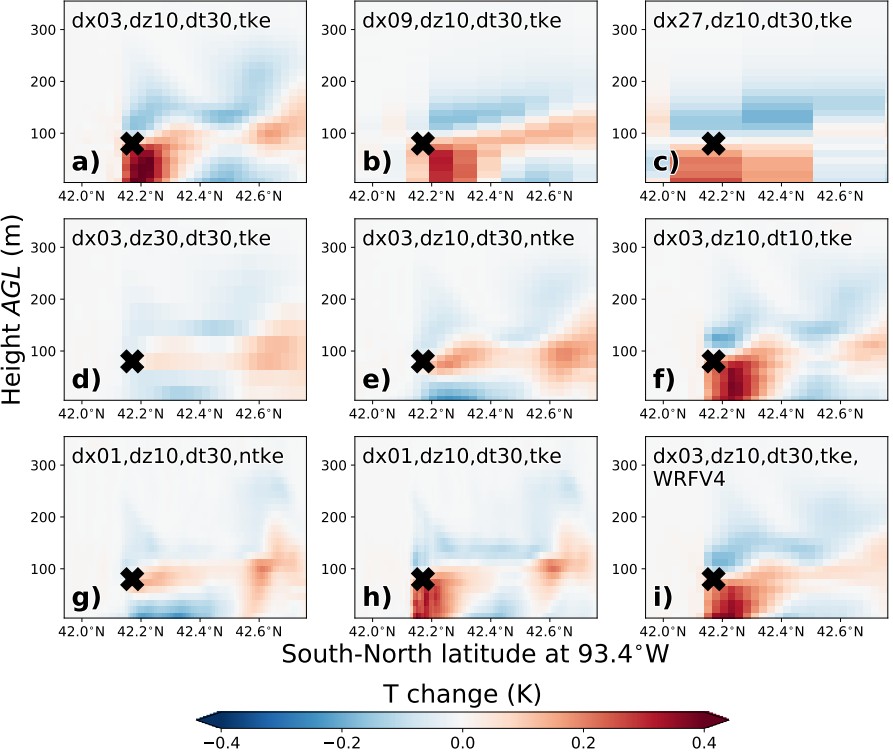

**Figure 11.** Vertical cross sections of temperature (T) changes resulting from the presence of the wind farm in the tested simulations on Aug 26 0200 UTC (Aug 25 2100 LT). The median location and hub height of the wind farm is denoted by the black X. Location of this slice is denoted by the dashed line in Fig. 10.

and turbine-generated turbulence term, as a too-coarse (i.e., 30 m) vertical grid or a lack of additional turbine turbulence fails to simulate enough wind shear to vertically mix warm inversion air to the surface, resulting in an incorrect surface cooling signal overnight.

Out of the four horizontal resolutions tested, the finer-grid (1 km and 3 km) configurations produce a more robust representation of wind farm wakes than the coarser grids (9 km and 27 km). The finer grid spacing allows stronger wake impacts both in the wind speed deficit and surface warming to develop over a more localized region, better matching observations (e.g., Siedersleben et al., 2018a). Coarser horizontal grids (>9 km) have been chosen in recent work (e.g., Vautard et al., 2014; Miller and Keith, 2018) for long-term or spatially large simulations because of the savings in computational expenses (see Table 2). However, such simulations imply a broader region of impact than is realistic (Fig. 6, 7, 8, 9). Configurations on the order of a few kilometers in the innermost turbine-containing domain are thus recommended. Close similarities between the 1-km and 3-km configurations suggest the user can confidently produce accurate waking with a 3-km WRF WFP grid spacing, refining to 1 km if the computation resources are available or if terrain complexity indicates that finer resolution is required.

The choice of vertical resolution significantly impacts WRF WFP solutions of wake effects, especially in the representation of nocturnal near-surface warming around the wind farm (Fig. 10d, 11d, 12b, 13, 14). A 30-m vertical grid consistently produces more incorrect cooling signals overnight than the baseline simulation with a 10-m grid. Observed surface warming in

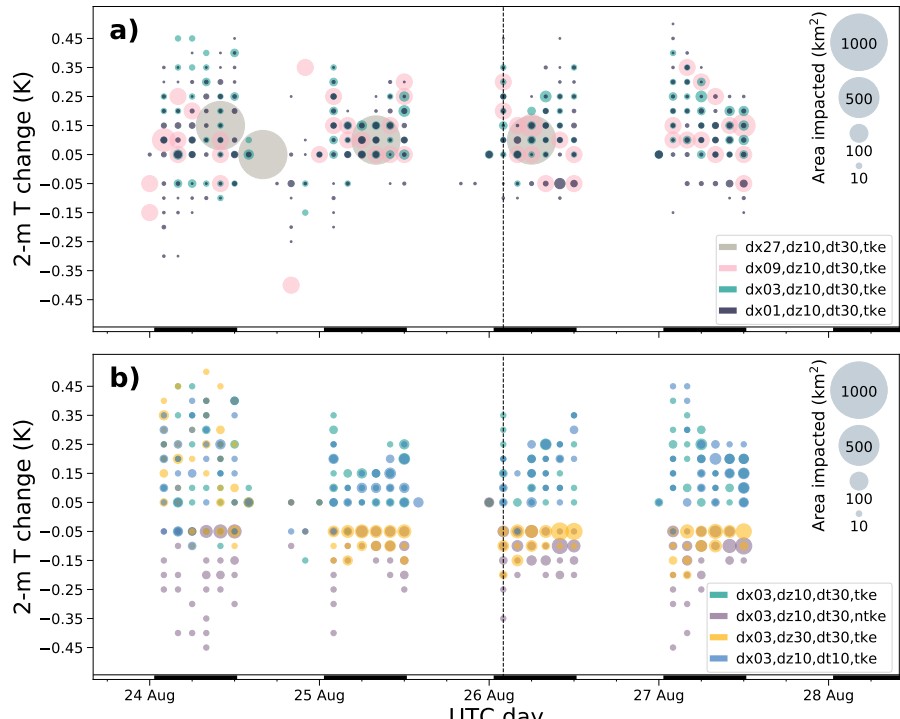

**Figure 12.** Time series of area impacted by the wake-induced, 2-m temperature changes as predicted by the tested simulations, plotted every 2 hours throughout the period. The size of the dots represents the spatial coverage of their respective magnitude of impact (scale denoted in top right), with each dot colored based on its configuration. The configurations are divided into groups that (a) vary horizontal resolution and (b) hold horizontal resolution constant while varying other model settings. The time chosen for the qualitative analysis (Aug 26 0200 UTC) in Fig. 6 is denoted by the black dashed line. Black and white bar at bottom denotes post-sunrise (white) and post-sunset (black).

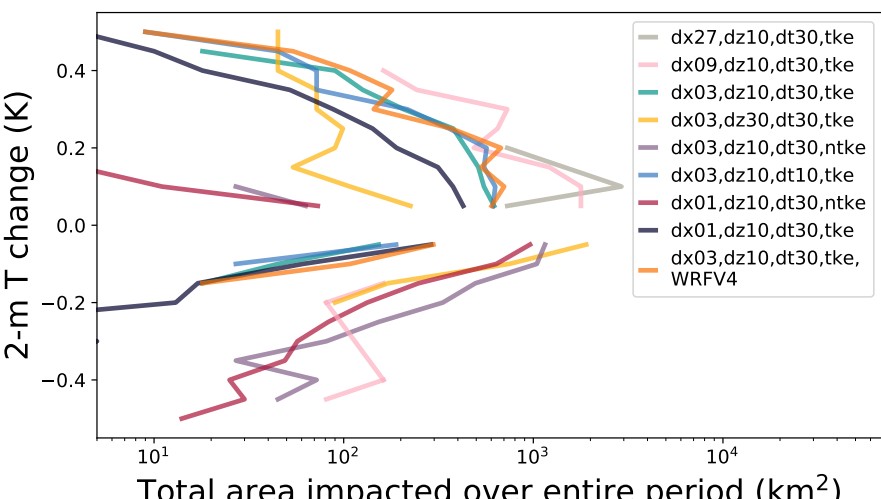

**Figure 13.** Total area impacted by the wake-induced 2-m temperature change as predicted by the tested simulations, plotted at different magnitudes of impact and integrated in time across the entire period. Each line is colored based on its configuration. Areas of warming are summed separately from areas of cooling.

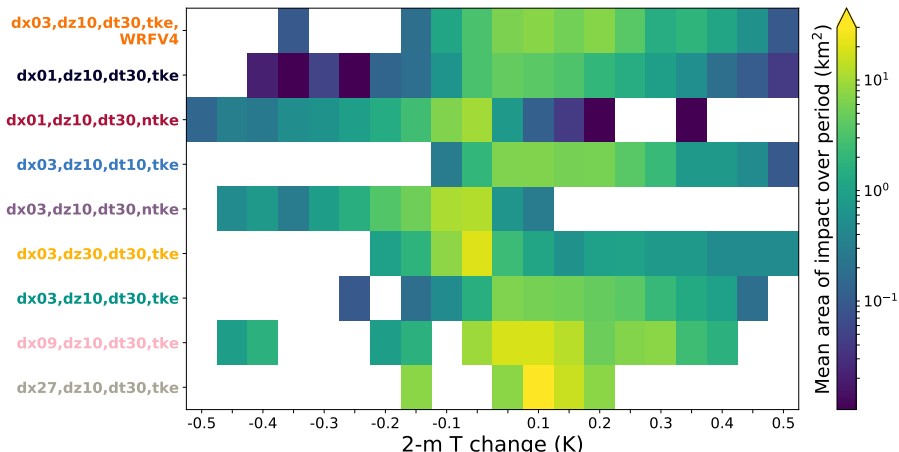

**Figure 14.** Average area affected by each 2-m temperature change (columns) over the entire simulation period for each simulation tested (rows). Yellow squares indicate larger areas of impact, with empty (white) squares indicating a lack of occurrence for that particular magnitude of impact.

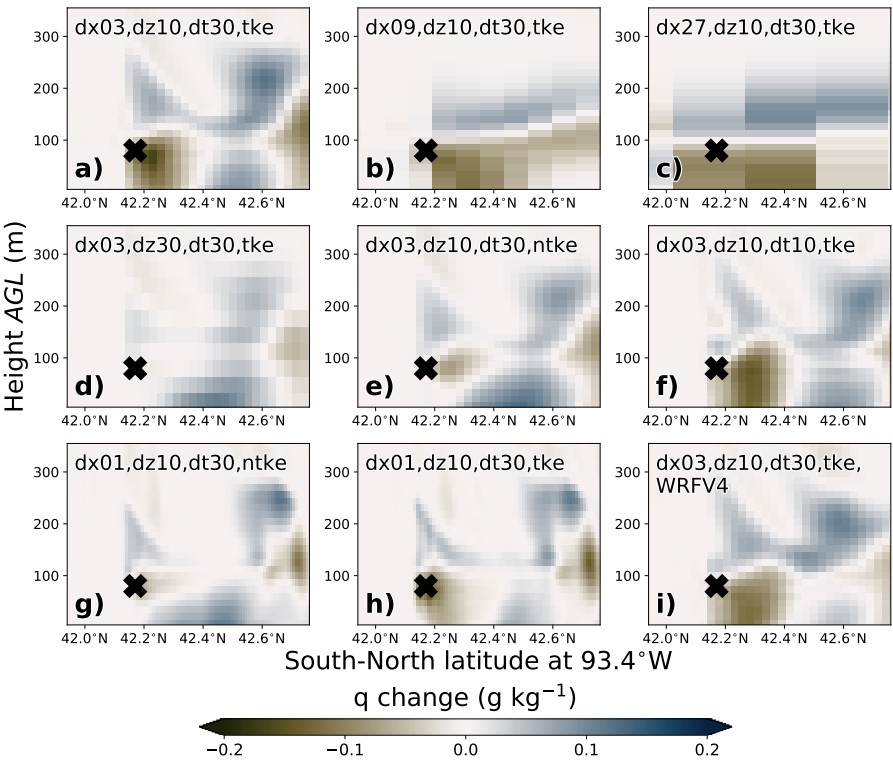

**Figure 15.** Vertical cross sections of water vapor mixing ratio (q) changes resulting from the presence of the wind farm in the tested simulations on Aug 26 0200 UTC (Aug 25 2100 LT). The median location and hub height of the wind farm is denoted by the black X. Location of this slice is denoted by the dashed line in Fig. 10.

the vicinity of wind farms occurs because of the turbine-generated turbulence mixing down warm air from above the nighttime inversion to the surface (Baidya Roy, 2004; Baidya Roy and Traiteur, 2010; Zhou et al., 2012; Rajewski et al., 2013). A sufficiently refined vertical grid is thus necessary to resolve this downward mixing, and a 30-m grid is inadequate. These findings

support prior work by Lee and Lundquist (2017a), who also concluded that a finer (∼12 m) vertical grid is favorable for the separate purpose of producing more accurate WRF WFP solutions of the winds and power production than a coarser (∼22 m) grid. However, coarse (>20 m) vertical resolutions have been employed in other past WRF WFP studies, possibly artificially constraining the temperature signal (e.g., occurrence of both negative and positive temperature signals seen in Vautard et al. 5 (2014)).

In addition, the turbine-induced turbulence option has similar impacts on the WRF WFP wake solution as the vertical resolution. When enabled, this turbulence option within the WFP adds an additional source of TKE within turbine-containing grid cells derived from the difference between the turbine thrust and power coefficients. Without this added TKE source, WRF wind farm wake turbulence only develops because of the wind shear that arises out of the momentum deficit aloft in 10 the wind farm wake. However, this shear-induced mixing is insufficient, as configurations without the added turbine-TKE option consistently produce inaccurate nocturnal cooling signals at the surface immediately beneath the wind farm (Fig. 10e,g 11e,g 12b, 13, 14). Such surface cooling implies that insufficient mixing is occurring within WRF WFP, making it unable to bring warm inversion air to the surface. This hypothesis is corroborated by Xia et al. (2019), who demonstrated that the WFP turbulence option is responsible for the surface warming signal through the enhancement of vertical mixing. As such, 15 the WFP turbine TKE option, in addition to sufficiently refined vertical and horizontal grid resolutions (∼10 m and ∼3 km, respectively), is required to represent wind farm wakes accurately.

## 5   Conclusions

As wind energy continues to rapidly develop, the Wind Farm Parameterization (WFP) within the Weather Research and Fore-casting (WRF) model provides a means for simulating wind farms and their large-scale wake effects. However, little guidance currently exists for choice in model settings to produce the most accurate solution of wakes. Herein, we assess the sensitivity of the WRF WFP to model configuration to provide recommended settings for simulating wind farm wakes effects.

We select Aug 24-27 of the 2013 Crop Wind Experiment (CWEX-13) field campaign as our case study because of the simple terrain, availability of observations, and consistent, nocturnal low-level jet occurrences without interference from large-scale synoptic meteorological events. We use measurements from a scanning lidar to first verify the ambient flow simulated by WRF 25 before implementing the WFP and varying the horizontal and vertical resolutions, turbine-generated turbulence, model version, and model time step settings to comprise the sensitivity analysis. Each model configuration simulates a real Iowa wind farm containing 200 1.5-MW turbines.

We isolate the impacts of WRF WFP settings on the two predominant meteorological effects of a wind farm wake, the hub-height wind speed deficit and the transient surface temperature increase arising out of downward mixing of the nocturnal 30 inversion. While the inclusion of the turbine-generated turbulence option in the WFP has little impact on the wind speed deficit solution, disabling it results in an inaccurate cooling signal beneath the wind farm. Similarly, a coarse (30 m) vertical resolution

has minimal impact on the representation of the wind deficit aloft, but impacts the surface temperature signal drastically by reversing the sign of the expected temperature impact. WRF WFP simulations thus require a ∼10-m low-level vertical grid as well as the turbine-turbulence option enabled to produce the wind shear necessary to vertically mix the inversion air and attain the expected surface warming and drying. Horizontal resolution affects both the wind speed deficit and surface warming: a too-coarse (>9 km) grid dilutes wake effect intensity over greater areas, while grids of 1 km or 3 km converge on similar depictions of the magnitude and spatial coverage of wake impacts and thus serve as our recommended horizontal grid choice.

In conclusion, the WRF WFP is sensitive to certain model settings, particularly (1) the horizontal resolution in producing accurate intensity and coverage of the wind speed deficit and surface temperature change; and (2) the vertical resolution and (3) turbine turbulence option in producing the correct surface warming signal. In order to obtain the most accurate representation of wind farm wakes, we suggest that users define a horizontal grid for the turbine-containing domain on the order of a few kilometers and a vertical grid near 10 m in the lowest ∼200 m. The inclusion of turbine-generated turbulence is also necessary. While model time step and model version had less impact on the wake solutions, these sensitivity evaluations should continue as WRF and the PBL schemes evolve.

This sensitivity study and subsequent model setting recommendations are derived from analysis of a single location and time period, and further analysis including a wider range of meteorological conditions or locations could be worthwhile, especially as wind energy develops more offshore and in complex terrain on land. While we predict the WFP wake solutions and model sensitivity in less-turbulent offshore environments will behave similarly to the simple terrain case studied herein, the more-turbulent flow over complex topography may alter how wakes are represented in the WRF WFP and thus impact the model sensitivity. The WFP is designed to work with the MYNN 2.5 level PBL scheme, so impacts that the choice in PBL scheme may have on the background meteorology and subsequent wake solution are not addressed. Furthermore, within-the-grid-cell turbine wake interactions are omitted in the WFP and not considered here. Future applications of the WRF WFP to investigate wind farm wake effects will have scientific and societal implications, so it is therefore important to consider model settings when designing simulations.

*Code and data availability.* The WRF-ARW model code (https://doi.org/doi:10.5065/D6MK6B4K) is publicly available at http://www2.mmm.ucar.edu/wrf/users/. This work uses the WRF-ARW model and the WRF Preprocessing System (WPS) version 3.8.1 (released on 12 August, 2016), and the wind farm parameterization is distributed therein. Initial and boundary conditions are provided by Era-Interim (Dee et al., 2011) available at https://rda.ucar.edu/datasets/ds627.0/. Topographic data are provided at a 30-s resolution from http://www2.mmm.ucar.edu/wrf/users/download/get_source.html. The PSU generic 1.5-MW turbine (Schmitz, 2012) is available at https://doi.org/10.13140/RG.2.2.22492.18567. The model namelists, wind turbine specifications, and parsed output data needed to recreate the figures and analysis are located at https://doi.org/10.5281/zenodo.3755282.

*Author contributions.* JKL and JMT conceived the research and designed the WRF simulations; JMT carried out the WRF simulations and wrote the manuscript with significant input from JKL.

*Competing interests.* The authors declare that they have no competing interests.

*Acknowledgements.* This work and JMT were supported by an NSF Graduate Research Fellowship under grant number 1144083. WRF simulations were conducted using the Extreme Science and Engineering Discovery Environment (XSEDE), which is supported by National Science Foundation grant number ACI1053575. JKL's effort was supported by an agreement with NREL under APUP UGA-0-41026-65.

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
