# Peer review of "Simulated wind farm wake sensitivity to configuration choices in the"

_Geoscientific Model Development, 2019_

## Short Comment (SC1) · 15 Feb 2020

General Comments: The paper is of great importance given the rapid increase in wind farm development. It is difficult to build a new wind farm without being influenced by wake effects from surrounding wind turbines. The open-source Weather Research and Forecast (WRF) model and coupled Wind Farm Parameterization (WFP) is a sophisticated numerical modeling tool for addressing the impact of wake effects. The paper's research findings provide a valuable WRF-WFP configuration template for both researchers and private industry to more accurately quantify the impact of wind farm

wake effects on regional wind resources. Increased modeling accuracy of potential wind power production reduces wind park cost uncertainty and contributes to the transition to more robust wind energy generation portfolios.

Minor comments: 1. The finest horizontal grid spacing included in the sensitivity study was 1 km. Why were finer resolutions not explored? Was it based on available computational resources or underlying physical limitations of the WFP? 2. One component of the sensitivity study was disabling TKE generation... was this accomplished via a WRF namelist setting or was the source code modified? 3. It may be useful to provide the eta levels used in the lower atmosphere to achieve 10 m vertical resolution in a Table as a reference for the modeling community. 4. Please discuss the choice of using 0.7 degree ERA-Interim initial / boundary condition data when higher resolution data sets (RUC/RAP/NAM) are available during the study period. My guess is the WRF parent domain (27 km) may be sufficiently north to be just outside the bounds of the mentioned data sets. 5. Ultimately, the accuracy of the WFP is limited by the WRF prediction skill of the background wind speed. The choice of model physics (PBL, LSM, surface layer) has been shown to have a significant impact on near surface wind prediction. Can you generally discuss how important choice of physics is relative to the WRF-WFP model configuration setup? As I understand it, the WFP only works with the MYNN 2.5 level PBL scheme, so I imagine that could be a limitation.

---

## Referee Comment (RC1) · Anonymous Referee #1 · 18 Feb 2020

General Comment:

The manuscript by Tomaszewski and Lundquist tests the sensitivitiy of the WRF models wind farm parametrisation to different configurations. Despite quite a number of studies that were recently released on this topic using the WRF model, the manuscript still provides some useful additional insight into the sensitivitiy of the wind farm parametrisation and gives an important overview on how to setup the model to get consistent and more accurate results. It is nicely written and also the figures are generally formatted well. However, I have one major comment and a number of minor comments and

thus recommend publication after dealing with major revisions.

Major Comment:

MYNN parametristaion changes: In the framework of the sensitivity studies that were conducted for the New European wind atlas different WRF versions were evaluated ([1] & [2]). From the analysis of theres results, a sensitivity towards different versions of the MYNN scheme was found. This scheme was modified in January 2016 [3] and the changes were reverted in 2018 [4], so WRF versions 3.7.X to 3.9.X are affected. As significant differences and biases in the wind were reported, the reviewer suggests to test the impact of this parameter by either changing it manually in the version run by the authors or comparing a run from a newer WRF version.

Minor Comments:

Page 1 - Line 7: Repetition of the word "undermining" - Please consider rewriting

Page 1 - Line 13: spinning turbines – Better: rotating?

Page 1 - Line 14: in situ observations – Better: in-situ observations?

Page 2 - Line 7: Reducing carbon dioxide emissions – wind energy itself does not reduce carbon dioxide emissions just emits significantly less (only during manufacturing) than burning fossil fuels thus emissions are reduced if wind energy is used instead of fossil energy sources. Please be more precise here.

Page 2 - Line 21: A number of LES studies are mentioned here, however, RANS and industry models (bottom up approach) are still commonly used for wind farm wake investigations as well. These models are however often lacking in parametrisations of meteorological effects sucht as atmospheric stability or large scale wind turnings. For completeness however, I suggest to add some RANS and industry model studies to the discussion here.

Page 4- Table 1: Near surface temperature impact – a warming effect of wind farms is

heavily discussed by wind energy sceptics at the moment, which are also sometimes taking some information from the context. That's why I suggest to add something like "due to redistribution of heat" to the "near surface temperature impact" in Table 1 just to make sure that the turbines are of course not warming but redistributing the heat in the lower atmosphere.

Page 5 – Line 31: ...Siedersleben et al. (2018b) show little sensitivity to the exact turbine power curve. - Please remove this sentence here as it can be misinterpreted. In case of the Siedersleben paper it was a special case where by chance the turbine power curves/thrust curves were similar. When investing this from the turbine technology development perspective it will make a very big difference if a turbine with a specific power of some 250 W/m$^2$ or some 400 W/m$^2$ ist extracting energy from the flow. This is particular true when using the Fitch model as the difference between CP and CT curves that determine the turbulence will also change.

Page 6 – Figure 1 : It is very hard to read the "1km" in the central most domain. Maybe one could remove the unit or just put this information to the caption?

Page 8 Line 2-4: Lee and Lundquist 2017a)... – I wonder how much this might be related to the changes of the MYNN model. (related to the major comment)

Page 18 – Line 10: ... meteorological wake effects – better: large scale wake effects Code and Data Availability: It would be extremely helpful for the reproduction of results if the namelists and also the final turbine specifications (txt tables) for the different runs could be provided. So please upload to e.g. zenodo and refer to them in the code and data availability section.

References:

[1] Witha et al. (2019): WRF model sensitivity studies and specifications for the NEWA mesoscale wind atlas production runs, Technical Report, 73 pages, doi: https://doi.org/10.5281/zenodo.2682604

[2] Hahmann et al. (2020): The Making of the New European Wind Atlas, Part 1: Model Sensitivity, submitted to Geoscientific Model Development

[3] https://github.com/wrf-model/WRF/commit/215265a928e1afd2e9f120833cad9a5d6f6b7563#diff-1a3f1e15af30c3ef38c4079db0d9c4d4

[4] https://github.com/wrf-model/WRF/commit/bddd7f449ff972a592e297baff4dda8153666d30#diff-1a3f1e15af30c3ef38c4079db0d9c4d4

---

## Referee Comment (RC2) · Christoph Knote (Referee) · 9 Mar 2020

This is an editor comment on the manuscript by Tomaszewski et al., after the second reviewer originally nominated has missed to send in a review despite several attempts.

Tomaszewski et al. report on the influence of model configuration choices on the wake created by a wind farm. These wakes are of serious concern for wind farm operators as they can potentially reduce power production at downstream wind turbines and thereby reduce revenues.

The paper is well written, clearly structured and the methods and results are presented in a concise manner. The thorough introduction is especially appreciated, as it easily allows the reader to put the current work into the broader context of the existing scientific literature.

I consider the Scientific Comment by Matthew Simpson to contain substantial and valid comments that the authors should treat as if it were a referee comment. This, in addition with the original first review provides a solid and sufficient peer review for this manuscript. Thereby, I suggest the authors respond to these two to end the discussion phase.

---

## Author Comment (AC1) · 20 Apr 2020

The comment was uploaded in the form of a supplement:
https://www.geosci-model-dev-discuss.net/gmd-2019-302/gmd-2019-302-AC1-
supplement.pdf

---

## Author Response (AR1)

**Tomaszewski et al. Responses to Reviewers**
**April 2020**

All reviewer comments appear in regular text below, while authors' responses appear in purple text. Line numbers referenced in the authors' responses refer to the revised document.

**Response to Anonymous Referee #1**

General Comment:
The manuscript by Tomaszewski and Lundquist tests the sensitivitiy of the WRF models wind farm parametrisation to different configurations. Despite quite a number of studies that were recently released on this topic using the WRF model, the manuscript still provides some useful additional insight into the sensitivitiy of the wind farm parametrisation and gives an important overview on how to setup the model to get consistent and more accurate results. It is nicely written and also the figures are generally formatted well. However, I have one major comment and a number of minor comments and thus recommend publication after dealing with major revisions.

Major Comment:
MYNN parametristaion changes: In the framework of the sensitivity studies that were conducted for the New European wind atlas different WRF versions were evaluated ([1] & [2]). From the analysis of theres results, a sensitivity towards different versions of the MYNN scheme was found. This scheme was modified in January 2016 [3] and the changes were reverted in 2018 [4], so WRF versions 3.7.X to 3.9.X are affected. As significant differences and biases in the wind were reported, the reviewer suggests to test the impact of this parameter by either changing it manually in the version run by the authors or comparing a run from a newer WRF version.

Thank you for your thoughtful review and very helpful point regarding changes to PBL schemes. We agree the study would be improved with a discussion of the sensitivity to WRF model version. Therefore, we have added a new case simulated with WRF version 4.0 to consider the MYNN and other changes. Accordingly, we have updated Figures 6, 8, 10, 11, 13, and 15, as well as included the WRF V4 test in newly added Figures 9 and 14, discussed in more detail at the end of this response. We found this sensitivity to the newer version of WRF to be subtle. Corresponding discussion has been added to the manuscript in the following places:

Section 1, page 4, line 10:

"Sensitivity studies conducted for the New European Wind Atlas find a sensitivity to modifications to the MYNN scheme in different WRF versions (Witha et al., 2019; Hahmann et al., 2020). The MYNN scheme within WRF Versions 3.7.X to 3.9.X differs from 4.X most notably in the drag coefficient parameterization in the surface layer subroutine and the mixing length

formulation, leading to differences in the wind that could impact wake studies (Olson et al., 2016)."

Section 2.2, page 6, line 2:

"We conduct all simulations but one in our sensitivity study with version 3.8.1 of the Advanced Research WRF (ARW) model (Skamarock and Klemp, 2008). While model time step, vertical resolution, horizontal resolution (and thus domain size), and model version are among the model settings varied to test sensitivity, several model options are kept consistent across all simulations based on previous studies of this time period…."

Section 2.2, page 6, line 27:

"We then vary the horizontal resolution (*dx*), vertical resolution (*dz*), time step(dt), turbulence option (*tke* or *ntke*), and WRF model version, 3.8.1 vs 4.0 (*V4*), about this baseline configuration to make up our sensitivity test (Table 2)."

Section 2.2, page 7, line 7:

"Finally, we assess sensitivity to WRF version by running a simulation with the same configuration as the baseline (*dx03_dz10_dt30_tke*) with version 4.0 of WRF."

Section 3.2.1, page 10, line 11:

"Using WRF version 4.0 (Fig. 6i) also creates subtle differences in the shape of the far wake and in the magnitude of the deficit in the near wake."

Section 3.2.1, page 11, line 9:

"The newer version of WRF predicted a similar time series of deficits as the baseline (green) but was omitted from Fig. 7 to reduce clutter."

Section 3.2.1, page 11, line 20:

"The baseline case run with version 3.8.1 and the case run with version 4.0 (green vs. orange) predict similar total waking impacts over the period, deviating most (~100 m$^2$) at the 1.8 m s$^{-1}$ deficit (Fig. 8)."

Section 3.2.1, page 11, line 26:

"Subtle sensitivity exists to the model time step and version, most apparent in the average areas impacted by the strongest deficits, i.e., 1.8 and 2.0 m s$^{-1}$ (Fig. 9)."

Section 3.2.2, page 12, line 6:

"Similarly, reducing the time step or using WRF version 4.0 has little impact on the model solution of temperature changes based on snapshots from Fig. 10f,i and 11f,i."

Section 3.2.2, page 13, line 18:

"Reducing model time step (blue) or using version 4.0 of WRF (orange) again has little impact on the overall prediction of temperature impact coverage compared to the baseline (green) (Fig. 13, 14)."

Section 4, page 14, line 7:

"Settings tested include horizontal and vertical grid spacing, model time step, model version, and inclusion of turbine-generated turbulence."

Section 5, page 20, line 31:

"We use measurements from a scanning lidar to first verify the ambient flow simulated by WRF before implementing the WFP and varying the horizontal and vertical resolutions, turbine-generated turbulence, model version, and model time step settings to comprise the sensitivity analysis."

Section 5, page 21, line 19:

"While model time step and model version had less impact on the wake solutions, these sensitivity evaluations should continue as WRF and the PBL schemes evolve."

An example of an updated figure is below, with the orange line serving as the WRF 4.0 analogue to the baseline configuration ran with WRF 3.8.1 (green line). Only subtle differences emerge.

[Figure]

**Figure 13.** Total area impacted by the wake-induced 2-m temperature change as predicted by the tested simulations, plotted at different magnitudes of impact and integrated in time across the entire period. Each line is colored based on its configuration. Areas of warming are summed separately from areas of cooling.

Minor Comments:
Page 1 - Line 7: Repetition of the word "undermining" - Please consider rewriting

We agree with rewriting the sentence on page 1, line 7 so that it now says:

"These wakes can undermine power production at downwind turbines, adversely impacting revenue."

Page 1 - Line 13: spinning turbines – Better: rotating?

We replaced the word "spinning" with the more accurate "rotating" on page 1, line 13.

Page 1 - Line 14: in situ observations – Better: in-situ observations?

We rewrote the sentence to say on page 1, line 14 to say:

"Here we compare simulated wind farm wakes produced by varying model configurations with meteorological observations near a land-based wind farm in flat terrain over several diurnal cycles."

Page 2 - Line 7: Reducing carbon dioxide emissions – wind energy itself does not reduce carbon dioxide emissions just emits significantly less (only during manufacturing) than burning fossil fuels thus emissions are reduced if wind energy is used instead of fossil energy sources. Please be more precise here.

We agree with improving the precision of the message here and have updated page 2, line 8 to say:

"Wind energy is growing rapidly to meet increasing energy demands with lower-carbon electricity sources."

Page 2 - Line 21: A number of LES studies are mentioned here, however, RANS and industry models (bottom up approach) are still commonly used for wind farm wake investigations as well. These models are however often lacking in parametrisations of meteorological effects sucht as atmospheric stability or large scale wind turnings. For completeness however, I suggest to add some RANS and industry model studies to the discussion here.

We agree that the discussion is improved with the addition of RANS and industry models and have updated page 2, line 24 to say:

"Reynolds-averaged Navier-Stokes (RANS) approximations (Cabezón et al., 2011; Tian et al., 2014; Göçmen et al., 2016; Astolfi et al., 2018; Iungo et al., 2018) and industry flow models (e.g., FLOw Redirection and Induction in Steady State (FLORIS; NREL, 2019) and Wind Farm Simulator (WFSim; Boersma et al., 2016)) are commonly used for lower-cost wind farm wake investigations (Beaucage et al. 2012). However, these models are often limited in parameterizations of meteorological effects such as atmospheric stability or large-scale wind patterns."

Page 4- Table 1: Near surface temperature impact – a warming effect of wind farms is heavily discussed by wind energy sceptics at the moment, which are also sometimes taking some information from the context. That's why I suggest to add something like "due to redistribution of heat" to the "near surface temperature impact" in Table 1 just to make sure that the turbines are of course not warming but redistributing the heat in the lower atmosphere.

The authors agree with the need to provide more context and have amended Table 1 on page 4 to say:

 "Near-surface T impact due to vertical redistribution of heat"

Page 5 – Line 31: ...Siedersleben et al. (2018b) show little sensitivity to the exact turbine power curve. - Please remove this sentence here as it can be misinterpreted. In case of the Siedersleben paper it was a special case where by chance the turbine power curves/thrust curves were similar. When investing this from the turbine technology development perspective it will make a very big difference if a turbine with a specific power of some 250 W/m2 or some 400 W/m2 ist extracting energy from the flow. This is particular true when using the Fitch model as the difference between CP and CT curves that determine the turbulence will also change.

The reviewer makes an important distinction here, and we have amended the text on page 6, line 18 to include a warning:

"This turbine model closely matches those installed in the wind farm present at the CWEX site, and Siedersleben et al. (2018b) show little sensitivity to the exact turbine power curve for similar turbines. For turbines with substantially different ratings, the exact power curves should be used."

Page 6 – Figure 1 : It is very hard to read the "1km" in the central most domain. Maybe one could remove the unit or just put this information to the caption?

We updated Fig. 1 with larger text and also added the information to the caption on page 7.

Page 8 Line 2-4: Lee and Lundquist 2017a)... – I wonder how much this might be related to the changes of the MYNN model. (related to the major comment)

Lee and Lundquist (2017a) also use version 3.8.1 as was used throughout most of this study. We did evaluate the differences with using WRF 4.0 as detailed in the earlier response to the major comment.

Page 18 – Line 10: ... meteorological wake effects – better: large scale wake effects

We replaced "meteorological" with "large-scale" on page 20, line 25.

Code and Data Availability: It would be extremely helpful for the reproduction of results if the namelists and also the final turbine specifications (txt tables) for the different runs could be provided. So please upload to e.g. zenodo and refer to them in the code and data availability section.

Thank you for the suggestion. We have uploaded the namelists and turbine text files to Zenodo and included the link to them in the Code and Data Availability section on page 22, line 6:

"The model namelists, wind turbine specifications, and parsed output data needed to recreate the figures and analysis are located at https://doi.org/10.5281/zenodo.3755282."

**Response to Scientific Comment from Matthew Simpson**

General Comments: The paper is of great importance given the rapid increase in wind farm development. It is difficult to build a new wind farm without being influenced by wake effects from surrounding wind turbines. The open-source Weather Research and Forecast (WRF) model and coupled Wind Farm Parameterization (WFP) is a sophisticated numerical modeling tool for addressing the impact of wake effects. The paper's research findings provide a valuable WRF-WFP configuration template for both researchers and private industry to more accurately quantify the impact of wind farm wake effects on regional wind resources. Increased modeling accuracy of potential wind power production reduces wind park cost uncertainty and contributes to the transition to more robust wind energy generation portfolios.

Thank you for your thoughtful review and helpful comments.

Minor comments:

1. The finest horizontal grid spacing included in the sensitivity study was 1 km. Why were finer resolutions not explored? Was it based on available computational resources or underlying physical limitations of the WFP?

When using a mesoscale model like WRF, we generally try to avoid operating at scales finer than 1 km to avoid complications with physical assumptions breaking down in a zone known as the *Terra Incognita*. We updated the text on page 7, line 2 to give references to this phenomenon and to justify simulating no finer than 1 km:

"Our finest domain tested is 1 km to avoid issues with the Terra Incognita (Wyngaard, 2004; Ching et al., 2014; Zhou et al.,2014; Doubrawa and Muñoz-Esparza, 2020)."

2. One component of the sensitivity study was disabling TKE generation... was this accomplished via a WRF namelist setting or was the source code modified?

Disabling the TKE generation is done by commenting out a line in the Wind Farm Parameterization code. We agree more detail is needed in the text for clarity and have including the following message in page 7, line 5:

"Disabling the TKE generation is done by commenting out line 226 (the qke(i,k,j) calculation) in module_wind_fitch.F and recompiling WRF."

3. It may be useful to provide the eta levels used in the lower atmosphere to achieve 10 m vertical resolution in a Table as a reference for the modeling community.

The authors agree that eta levels would be useful for reproducibility. These are included in the namelist files linked in the Code and Data Availability at https://doi.org/10.5281/zenodo.3755282.

4. Please discuss the choice of using 0.7 degree ERA-Interim initial / boundary condition data when higher resolution data sets (RUC/RAP/NAM) are available during the study period. My guess is the WRF parent domain (27 km) may be sufficiently north to be just outside the bounds of the mentioned data sets.

Previous studies have compared ERA-I to other reanalysis datasets and found better performance with ERA-I. We added this important reasoning to page 6, line 6:

"The 0.7∘ERA-Interim (ECMWF, 2009; Dee et al., 2011) data set provides initial and boundary conditions for all model runs, chosen for its better performance over other reanalysis data sets (Lee and Lundquist,2017a; Hahmann et al., 2020)."

5. Ultimately, the accuracy of the WFP is limited by the WRF prediction skill of the background wind speed. The choice of model physics (PBL, LSM, surface layer) has been shown to have a significant impact on near surface wind prediction. Can you generally discuss how important choice of physics is relative to the WRF-WFP model configuration setup? As I understand it, the WFP only works with the MYNN 2.5 level PBL scheme, so I imagine that could be a limitation.

The reviewer brings up an excellent point. The WFP is indeed designed to work with the MYNN2.5 PBL scheme, so we currently cannot explore how the choice in PBL scheme may impact the background meteorology and subsequent wake calculations. We acknowledge this limitation in our concluding discussion of limitations and future work in page 21, line 26:

"The WFP is designed to work with the MYNN 2.5 level PBL scheme, so impacts that the choice in PBL scheme may have on the background meteorology and subsequent wake solution are not addressed."

**Additional note from authors**

Scientific conversations following a conference presentation of this work sparked an idea for new figures to compare the average wake effects predicted by the different simulation cases. We thus wish to incorporate these figures (shown on the next page) into the results to corroborate the other analyses in the paper. Discussion for the new Fig. 9 is on page 11, line 22:

[revised manuscript text omitted]